# Improving Multiclass Classification of Fake News Using BERT-Based Models and ChatGPT-Augmented Data

Elena Shushkevich [1], Mikhail Alexandrov [2] and John Cardiff [1,*]

1 Tallaght Campus, Technological University of Dublin, D24 FKT9 Dublin, Ireland; elena.n.shushkevich@gmail.com
2 Department of French and Romance Philology, Autonomous University of Barcelona, 08193 Bellaterra, Spain; malexandrov.uab@gmail.com
* Correspondence: john.cardiff@tudublin.ie

**Abstract:** Given the widespread accessibility of content creation and sharing, false information proliferation is a growing concern. Researchers typically tackle fake news detection (FND) in specific topics using binary classification. Our study addresses a more practical FND scenario, analyzing a corpus with unknown topics through multiclass classification, encompassing true, false, partially false, and other categories. Our contribution involves: (1) exploring three BERT-based models— SBERT, RoBERTa, and mBERT; (2) enhancing results via ChatGPT-generated artificial data for class balance; and (3) improving outcomes using a two-step binary classification procedure. Our focus is on the CheckThat! Lab dataset from CLEF-2022. Our experimental results demonstrate a superior performance compared to existing achievements but FND's practical use needs improvement within the current state-of-the-art.

**Keywords:** fake news detection; transformers; ChatGPT; mBERT; SBERT; XLM-RoBERTa; multiclass classification

## 1. Introduction

The widespread use of news portals and social media platforms has resulted in an influx of users seeking easy access to up-to-date content. Unfortunately, this surge in popularity has also led to a proliferation of fake news. Detecting and combating fake news has become a paramount challenge in modern human communication. Fake news covers a wide range of topics, such as politics, COVID-19, and ecology, each with its distinct characteristics. Currently, the problem of fake news detection typically revolves around binary classification, distinguishing fake news from real news within a specific topic. Modern models, particularly those employing transformers, have shown promising results in this regard.

However, in real-world scenarios, fake news is rarely confined to a single topic. It is not uncommon to encounter articles containing both political and COVID-19-related fake news, or a news article that combines politics and ecology topics. Furthermore, many instances of fake news incorporate a mixture of false and true information, capitalizing on the credibility associated with elements of truth. This blending of fact and fiction may be deliberate or unintentional on the part of the fake news creators.

Given these complexities, our focus lies in addressing the detection of nonspecific fake news in social media and implementing multi-class classification. This approach brings us closer to the real-world context, where fake news lacks a specific topic and requires classification into multiple classes.

We have three main objectives for this work:

- We will explore the potential of multi-class fake news detection, covering the categories false (fake news), true (real news), partially false (articles that contains both false and true statements), and others (articles without clear classification evidence).

We will utilize a dataset mirroring real-world content, spanning topics like politics, health, crimes, environment, and more. This will involve leveraging state-of-the-art transformers. To enhance the quality of fake news detection, we will balance and augment the existing dataset of fake news using messages generated by ChatGPT [1].

- We will propose novel classifiers that combine different classes to improve the accuracy of fake news detection. Additionally, we will compare this approach with the improvements gained from utilizing ChatGPT-generated news.
- By pursuing these goals, we aim to advance the field of fake news detection, particularly in the realm of social media, and enhance the accuracy and effectiveness of detecting fake news across multiple classes.

In this paper, we structure our work as follows: the first section introduces the problem and objectives of our study, followed by Section 2, which presents a comprehensive literature review in the area. In Section 3, we provide details about the datasets we utilized, while Section 4 covers the models employed and the experimental results we obtained. Section 5 focuses on the experiments conducted with the novel approach of AI-generated news. Additionally, in Section 6, we present an innovative approach using combined classes to address the multiclass fake news detection problem, along with the results obtained through this approach. Finally, in Section 7, we present our conclusions and engage in discussions about future work prospects.

## 2. Literature Review

The issue of fake news detection encompasses a broad scope, raising important questions and offering various approaches that can captivate researchers. In this section, we explore existing techniques for identifying fake news and delve into the notable contributions of other researchers who have achieved promising results in this domain. While our work focuses on multi-class fake news detection, it is essential to first grasp the fundamentals of fake news classification as a whole in order to implement multi-class classification effectively. To facilitate comparison and analysis, we have categorized fake news detection approaches into three primary groups: classical machine learning, neural networks, and other innovative methods. Additionally, we will examine studies specifically dedicated to the multi-class classification of fake news.

### 2.1. Classical Machine Learning Approach for Fake News Detection

Classical machine learning algorithms are widely employed for fake news detection, particularly in binary classification tasks. Previous studies have demonstrated their effectiveness in achieving satisfactory results.

In their study [2,3], the researchers aimed to detect COVID-19-related fake news using a small dataset of 1000 instances. Comparing various algorithms such as Logistic Regression, Support Vector Machine, Gradient Boosting, and Random Forest, Support Vector Machine and Random Forest achieved the highest performance with a 69% micro-F1 score. These studies can prove beneficial for researchers working with limited datasets. In [4], the authors compared algorithms on a large COVID-19 dataset from multiple social media platforms, with the SVM model achieving the highest F1-score of 0.93.

In their study [5], the authors conducted an analysis of fake news pertaining to the COVID-19 pandemic. To gather data, they collected a substantial dataset from various social media platforms, encompassing Twitter, email, mobile applications (such as WhatsApp), and Facebook. This data collection spanned a duration of four months, from March 2020 to June 2020. The researchers employed the traditional K-Nearest Neighbor algorithm and achieved F1-scores of 0.79 and 0.91 for the months of March and June, respectively.

### 2.2. Neural Networks for Fake News Detection

Within the realm of fake news detection, researchers have explored several intriguing linguistic models, with one of the most prominent being BERT (Bidirectional Encoder

Representations from Transformers) [6]. BERT has exhibited a remarkable performance across various natural language processing tasks, including text classification [7–9].

RoBERTa, an enhanced version of BERT, exhibits robustness but demands extensive training on larger datasets [10]. Another notable model is Hierarchical Attention Networks (HAN), which utilizes LSTM architecture. HAN operates through a hierarchical structure involving word-level and sentence-level attention mechanisms [11]. HAN has shown promising results, particularly in the realm of political fact-checking [12].

A modified version of BERT called DistilBERT has gained attention for its reduced size and enhanced speed, providing a 60% acceleration rate [13]. Additionally, the CT-BERT (COVID-Twitter-BERT) model, derived from DistilBERT, has specifically demonstrated impressive results in detecting fake news within Twitter messages [14]. In another research effort [15], the authors employed a widely recognized Convolutional Neural Network (CNN) to identify fake news related to COVID-19 in the LIAR dataset. Utilizing binary classification, they achieved an accuracy of 0.46.

In their study, the authors of [16] developed an ensemble of linguistic models including XLNet, RoBERTa, XLM-RoBERTa, DeBERTa, ERNIE 2.0, and ELECTRA to tackle the task of fake news detection. Impressively, they achieved an outstanding F1-score of 0.98.

Using the advanced BERT natural language processing model, the authors [17] secured the top spot in the fiercely competitive IberLEF 2021 shared task [18,19] (the model obtained an accuracy of 0.69), which aimed to detect fake news in Spanish.

The Constraint@AAAI2021—COVID19 Fake News Detection competition focused on detecting fake news. The challenge involved binary classification in English and Hindi, with the English dataset containing 10,700 messages. Out of 166 teams, 114 surpassed the baseline F1-score of 93%. The winning g2tmn team [20] achieved an impressive 98.69% F1-score using an ensemble of three pretrained CT-BERT models. The saradhix team [21] achieved a close second with 98.65% F1-score, employing classical machine learning methods and Transformers. The third-ranked xiangyangli team [22] obtained a 98.60% F1-score, utilizing Text Transformers and a Pseudo Label Algorithm for data augmentation. Notably, the best-performing models were ensemble-based Text Transformers, with fine-tuning being a critical step. Preprocessing techniques had a minimal impact in this context.

*2.3. Additional Features and Approaches for Fake News Detection*

In the field of natural language processing, various features and approaches have proven valuable for detecting fake news. These include GloVe for word representations [23], FastText classifiers [24], adversarial training [25], and tax2vec semantic vectorization [26]. GloVe transforms words into meaningful vectors, while FastText harnesses subword information for improved context awareness. Adversarial training introduces resilience, and tax2vec employs semantic vectorization with domain-specific knowledge.

The usage of fact-checking datasets is a crucial tool in combating fake news. These datasets provide valuable collections of labeled claims and verified truths, enabling machine learning models to make informed assessments about the accuracy of news articles and information circulating online. An example of such a dataset is FakeCOVID [27], the first multilingual dataset for COVID-19 fake news. FakeCOVID encompasses news from 150 countries in 40 languages, including 5182 fact-checked articles for COVID-19 collected between 4 January 2020 and 15 May 2020. The dataset draws from 92 different fact-checking websites to ensure reliability. Moreover, Breaking! [28] is a manually verified corpus of compelling fake and questionable news articles related to US politics, consisting of 679 articles in English categorized as false, partial truth, and opinions, collected during and before the 2016 US presidential election.

In the context of COVID-19, studies have examined the prevalence of fake news on different social media platforms. Twitter has been found to have a higher percentage of misinformation (19%) compared to Facebook (4%) and Reddit (7%) [29]. Additionally, research has explored the contribution of different topics in media publications, distinguishing

between reliable and unreliable information [30]. These additional features and approaches provide insights for more effective fake news detection and understanding its impact.

In summary of the comprehensive literature review conducted, it becomes evident that the most remarkable advancements in fake news detection, particularly in the domain of multiclass fake news classification, have been achieved through the utilization of neural networks. Notably, the most promising outcomes emerged when employing transformer-based models.

### 2.4. Multi-Class Classification of Fake News

Within the context of fake news detection, addressing multi-class classification scenarios where news articles encompass varying degrees of truthfulness becomes crucial. This approach recognizes that news data can span a spectrum from completely fake to completely real, with shades of truth and falsehood in between.

One significant study [31] focuses on classifying political materials from the LIAR dataset into five classes: false, barely-true, half-true, mostly-true, and true. Leveraging LSTM, the authors achieve an accuracy of 0.42 on a training set containing 10,200 texts and a test set of 1200 texts.

Another research effort [32] delves into the classification of fake news across six distinct classes: pants-fire, false, barely-true, half-true, mostly-true, and true. Employing SVM, Logistic Regression, Bi-LSTM, CNN, and Hybrid CNNs, the researchers achieve the highest accuracy of 0.27 using Hybrid CNNs.

The challenge of multi-class fake news detection garnered significant attention at the CLEF-2022 conference through the Shared Task 3—CheckThat! Lab [33]. Participants were tasked with classifying news articles in both English and German into four classes: true, partially true, false, or other. The "partially true" category represented news articles with a blend of true and false information, while the "other" category encompassed articles lacking sufficient evidence for classification. The provided dataset covered diverse topics.

The top-performing team in the English dataset achieved a macro-averaged F1-score of 0.34, utilizing a BERT-base-uncased model [34]. Although experiments with RoBERTa were conducted, the BERT model exhibited superior performance. A noteworthy ensemble of a Funnel Transformer and a Feed Forward Neural Network secured the second-best result, achieving a macro-averaged F1-score of 0.33 [35].

For the English–German cross-lingual task, the highest macro-averaged F1-score attained was 0.29, leveraging the BERT-large model [36]. Notably, the mDeBERTa model emerged as the second-best performer, achieving a macro-averaged F1-score of 0.23 [33].

It is important to highlight that the problem of multi-class classification in the context of fake news detection remains an area warranting further exploration. Even with sophisticated models and extensive training, achieving high accuracy in multi-class classification proves more challenging compared to binary classification scenarios.

The research on multi-class classification of fake news provides valuable insights into the complexities of categorizing news articles across multiple classes. It underscores the need for ongoing efforts to refine and advance classification techniques in order to effectively combat the spread of misinformation.

In summarizing our literature research, it becomes evident that transformer-based models, particularly those fine-tuned for specific cases, consistently exhibit the highest performance. Therefore, drawing from the substantial body of evidence, it is judicious to prioritize transformer models as our primary approach in this study. Their demonstrated capability to deliver superior results and effectively handle the intricacies of fake news classification reinforces their selection.

### 3. Datasets

In this section, we provide an overview of the datasets utilized for our experiments on multiclass fake news classification.

### 3.1. Basic Dataset for CheckThat-2022

For our experiments, we employ the CheckThat-2022 Task 3 dataset [33] as the primary dataset. We discussed this challenge in the previous section and highlighted the best results achieved on this dataset. Specifically, we focus on the English portion of the dataset.

The dataset, spanning 2010 to 2022, includes four labels:

- True: Articles with demonstrably true main claims.
- Partially False: Information that is not entirely accepted as true.
- False: Articles with untrue main claims (referred to as "fake news").
- Other: Articles lacking sufficient evidence for classification as true, false, or partially false.

This category encompasses disputed and unproven articles. Table 1 presents the dataset's statistics. Table 2 provides examples for each classification class.

**Table 1.** Statistics of the CheckThat-2022 dataset.

| | |
|---|---|
| Number of False messages | 893 (48% of all messages) |
| Number of True messages | 421 (22% of all messages) |
| Number of Partially false messages | 414 (22% of all messages) |
| Number of Other messages | 148 (8% of all messages) |
| Total number of messages | 1876 |
| Average length of a message | 731 words |

**Table 2.** Human generated news from the CheckThat-2022 dataset.

| Our Rating | Example |
|---|---|
| Other | Coronavirus horror: Volunteer in Oxford University's COVID-19 vaccine trial DIES |
| Partially False | Missouri politicians have made statements after the mass shooting late Sunday night in Las Vegas. In a tweet, Republican Senator Roy Blunt said he was saddened by the tragic loss of life and that his thoughts were with the families affected by "this horrific attack"... |
| False | It's no secret that Epstein and Schiff share a long history of perversion. Last year, it was revealed by the FBI that the two corresponded by email on several occasions... |
| True | WASHINGTON, D.C.—Ohio Treasurer Josh Mandel, a Republican candidate for U.S. Senate in 2012, on Friday released a copy of his delinquent personal financial disclosure report... |

The training dataset comprises 900 messages, the development dataset contains 364 messages, and the test dataset includes 612 messages for our experiments.

### 3.2. Dataset Augmentation Using ChatGPT

Given the relatively small and imbalanced nature of the CheckThat-2022 dataset, we sought to address these limitations by augmenting the dataset. To achieve this, we leveraged ChatGPT 4 [1], as there is a lack of real news datasets available specifically for multiclass fake news classification. Our aim was twofold: first, to explore the potential of AI in generating fake news and other categories, and second, to expand and balance the CheckThat-2022 dataset.

Using ChatGPT, we generated new messages that were designed to be similar but not identical to the original news articles in the CheckThat-2022 dataset. The graphical interpretation of the training dataset expansion using ChatGPT is presented in Figure During the generation of the false group of news using ChatGPT, it is worth mentioning that in certain cases, ChatGPT included warnings about the fabricated nature of the generated

news. However, we manually removed these warnings during our work. The examples of messages from the CheckThat-2022 dataset are presented in Table 2, and the examples of AI-generated news are presented in Table 3.

**Table 3.** News generated by GPT4 using the CheckThat-2022 Dataset.

| Our Rating | Example |
|---|---|
| Other | Pandemic tragedy: Participant in Moderna's COVID-19 vaccine research PASSES AWAY |
| Partially False | After the devastating terror attack in New York last weekend, Kansas lawmakers have expressed their distress. In a public message, GOP Senator Mark Davidson shared his deep grief regarding the terrible incident. . . |
| False | The rumor that Gates and Pelosi have a controversial past is far from hidden. Details emerged last year through an FBI announcement stating the two had been engaged in email correspondences, as many as 256 emails surfaced. . . |
| True | AUSTIN, TEXAS—Texas Secretary of State, Mark Walker, a Democratic contender for the U.S. Senate in 2013, on Tuesday revealed his overdue comprehensive financial disclosure statement, a 46-page document. . . |

This approach allowed us to create a balanced training dataset, increasing the number of messages for each of the four classes (false, true, partially false, other) to 600 news articles. The statistical overview of the augmented training dataset is presented in Table 4.

**Table 4.** Statistics of the training CheckThat-2022 dataset expanded by ChatGPT.

| Our Rating | CheckThat-2022 | GPT4 Generated |
|---|---|---|
| Other | 76 | 524 |
| Partially False | 217 | 383 |
| False | 465 | 135 |
| True | 142 | 458 |

Consequently, the training dataset saw a significant increase in the number of news articles, expanding from 900 to 2400. The visual representation of the updated training dataset can be found in Figure 1.

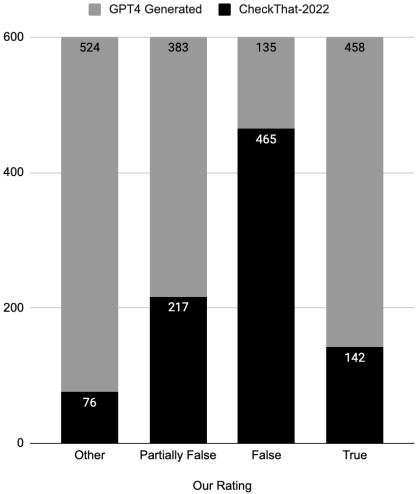

**Figure 1.** Percentage ratio of different messages in the training CheckThat-2022 dataset expanded using ChatGPT.

The graphical interpretation of the training dataset expansion using ChatGPT is presented in Figure 2.

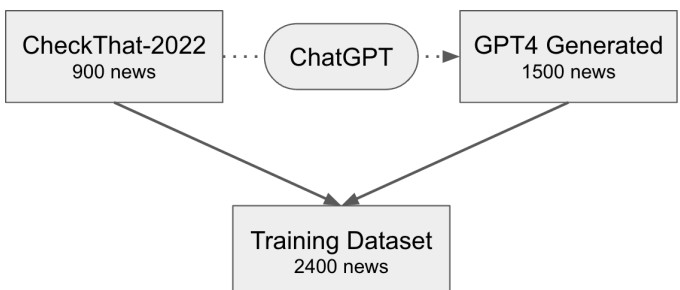

**Figure 2.** The training dataset expansion using ChatGPT.

## 4. Evaluation of BERT-Based Models

### 4.1. Selection of BERT-Based Models

In our study, we employed advanced transformer models, namely:

- mBERT [6]: This model, with 12 layers, 768 hidden units, and 110 M parameters, supports 104 languages. Pretrained on Wikipedia articles from the top 104 languages, it employs masked language modeling (MLM) for training. Our previous work has demonstrated the exceptional performance of BERT models in fake news detection, and mBERT can be utilized not only for English but also for multilingual experiments. To suit the requirements of our English dataset, we implemented a well-considered approach. This involved selecting a batch size of 8, which is optimal for managing computational resources effectively while ensuring thorough training. In addition, input sequences of 128 tokens were chosen to strike a balance between capturing context and maintaining computational efficiency. To prevent overfitting and enhance model generalization, a dropout rate of 0.5 was applied during training. These choices collectively contribute to a robust and comprehensive strategy for addressing our English dataset's specific characteristics.

- SBERT [37]: SBERT stands as an evolved rendition of BERT, thoughtfully designed to encompass a siamese structure and a triplet network configuration. This adaptation facilitates the creation of sentence embeddings imbued with semantic significance. These embeddings lend themselves to comparison via cosine similarity, a mechanism that accelerates processing speed while upholding the caliber of results comparable to BERT. Noteworthy is SBERT's multilingual support, encompassing a diverse range of over 100 languages, enabling the pursuit of cross-lingual and multilingual experimentation. Incorporating SBERT into our study showcases our commitment to harnessing cutting-edge models for enhanced outcomes. Our parameter choices for SBERT involved a batch size of 8, an input sequence length of 128 tokens, and a dropout rate of 0.3. These decisions are a testament to our dedication to fine-tune the model for optimal performance within our experimental framework.

- XLM-RoBERTa [38]: Our arsenal includes the XLM-RoBERTa model, an impressive cross-lingual sentence encoder. This powerhouse was trained using meticulously filtered CommonCrawl data across an extensive 100 languages, an endeavor that has culminated in its ability to attain state-of-the-art results across diverse cross-lingual understanding (XLU) benchmarks. An exceptional facet of this model is its pronounced potential for enabling multilingual classification tasks. In our pursuit of harnessing the capabilities of the XLM-RoBERTa model, we meticulously fine-tuned its parameters. This encompassed selecting a batch size of 16, strategically tailored to optimize computational resource utilization without compromising performance. To accommodate comprehensive context capture, input sequences of 256 tokens were opted for, a choice that ensures the model's ability to process meaningful chunks of information. Furthermore, a dropout rate of 0.3 was integrated into the training process,

serving to mitigate overfitting and enhance the model's generalization capabilities. This intricate interplay of configuration choices reflects our dedication to leveraging the XLM-RoBERTa model's potential for the advancement of multiclass classification tasks.

In order to maintain uniformity across our approach, we adopted a consistent learning rate of $1 \times 10^{-7}$ for all the models under scrutiny. Exploiting the potential of these BERT-based models, we conducted rigorous experiments using two distinct datasets. The first dataset was the English subset of the CheckThat-2022 dataset. Complementing this, we compiled a composite dataset comprising the English segment of CheckThat-2022 and messages meticulously generated via ChatGPT. Our innovative contribution lies in the exclusive utilization of AI-generated messages for the training phase, while the test dataset remained exclusively derived from the CheckThat-2022 dataset. All experimental procedures were executed on Amazon Azure, establishing a robust and standardized platform for our endeavors. This meticulous orchestration and configuration mirror our commitment to methodological precision, ensuring a robust foundation for our investigation's outcomes.

*4.2. Experimental Results with Basic Datasets*

We conducted experiments using three different transformer models, namely mBERT, SBERT, and XLM-RoBERTa, on the CheckThat-2022 dataset. The graphical interpretation of the experiments is presented in Figure 3.

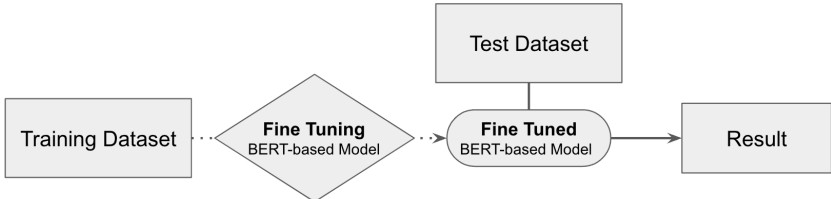

**Figure 3.** Experiments with BERT-based models.

The results of these experiments are summarized in Table 5.

**Table 5.** Experimental results on the CheckThat-2022 Dataset.

| Model | Accuracy | Macro F1-Score |
|---|---|---|
| mBERT | 0.34 | 0.23 |
| XLM-RoBERTa | 0.25 | 0.21 |
| SBERT | 0.31 | 0.20 |

Among these models, mBERT achieved the highest results, indicating its potential as the most promising model. To provide a more detailed analysis, we present the complete results obtained using the mBERT model in Table 6.

**Table 6.** Results of experiments with mBERT model.

| Class | Precision | Recall | Macro F1-Score |
|---|---|---|---|
| False | 0.52 | 0.43 | 0.47 |
| True | 0.34 | 0.32 | 0.33 |
| Partially false | 0.08 | 0.09 | 0.08 |
| Other | 0.02 | 0.06 | 0.03 |

As observed in Table 6, the classification results across different classes vary significantly. The classifier shows higher accuracy in predicting false messages compared to other classes. Our future experiments will focus on improving the classification performance for the remaining classes.

## 5. Experiments with the Expanded Dataset

As mentioned earlier, we explored the potential of enhancing the CheckThat-2022 dataset by incorporating news generated by ChatGPT. The results of experiments conducted on the expanded dataset using mBERT (the model that yielded the highest results in the initial experiments) are presented in Table 7.

**Table 7.** Results of experiments on the CheckThat-2022 + ChatGPT Generated News.

| Model | Accuracy | Macro F1-Score |
|---|---|---|
| mBERT | 0.32 | 0.25 |

The macro F1-score exhibited a 2% improvement compared to the results obtained using the training dataset without AI-generated news we presented in Table 5. A detailed breakdown of the classification results on the expanded dataset is provided in Table 8.

**Table 8.** Results of experiments with mBERT model on the expanded dataset.

| Class | Precision | Recall | Macro F1-Score |
|---|---|---|---|
| False | 0.52 | 0.30 | 0.38 |
| True | 0.36 | 0.41 | 0.38 |
| Partially false | 0.15 | 0.18 | 0.17 |
| Other | 0.04 | 0.16 | 0.06 |

The results obtained from the expanded dataset surpass those of the single CheckThat-2022 dataset, affirming our assumption that increasing the training data leads to improved classification outcomes in the context of multi-class fake news classification, although this improvement is not large. It is easy to see that the results for the true, partially false, and other classes witnessed improvements of 5%, 9%, and 3%, respectively, while the results for false news experienced a decline of 9%. This could be attributed to the limitations of ChatGPT in generating false news, potentially due to ethical considerations. Nonetheless, ChatGPT demonstrates the ability to generate true, partially false, and other categories of news, which can be leveraged to enhance the classification results.

## 6. Experiments with the Combined Classes

### 6.1. Combined Classes

As previously mentioned, achieving high-performance results in multi-class fake news classification on the CheckThat-2022 dataset remains a challenge. Therefore, our primary objective is to enhance the performance on this dataset. To address this, we propose an approach based on combining different classes to identify their intersection.

The concept behind this approach stems from the observation that binary classification of fake news often yields superior results compared to multi-class classification. With this in mind, we aim to convert our multi-class classification task into multiple binary classifications.

In the first step of our approach, we combine the true, partially false, and other news into a single joint class and perform binary classification against false news. Additionally, we combine false, partially false, and other news into another joint class and conduct binary classification against true news and the joint class. We repeat this process for two more binary classifications: partially false and the joint class, and other and the joint class. Each joint class consists of the corresponding classes of news.

In the second step, we assign labels to the messages in the test dataset based on the outcomes of the binary classifications. Messages labeled as false during binary classification are classified as false, those labeled as true are classified as true, partially false messages retain their label, and messages labeled as other are classified as other. Figure 4 illustrates the graphical representation of our approach: the two-step classification of fake news.

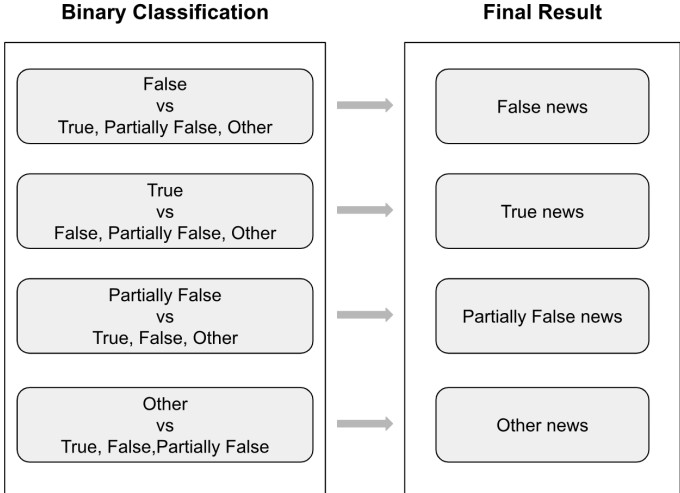

**Figure 4.** Two-step classification of fake news.

This final result is determined by identifying the class with the higher probability of belonging in cases where a message receives multiple labels. If a message does not receive any label, we assign the class indicated during the basic classification using mBERT from Section 4.

We believe that this two-step approach will contribute to improving the classification quality compared to a one-step multi-class classification.

### 6.2. Experiments with Combined Classes

Our experiments were conducted in two steps. Firstly, we performed four binary classifications, pitting each class against the combined set of the remaining classes. The results of these binary classifications are summarized in Table 9.

**Table 9.** Results for the binary classifications.

| Classification | Accuracy | Macro F1-Score |
|---|---|---|
| False vs. True, Partially False, Other | 0.51 | 0.45 |
| True vs. False, Partially False, Other | 0.65 | 0.41 |
| Partially False vs. True, False, Other | 0.81 | 0.57 |
| Other vs. True, False, Partially False | 0.85 | 0.49 |

While the accuracy for false news remains consistent in both binary and multi-class classification, the binary classification yields significantly better results for true, partially false, and other classes compared to multi-class classification. This confirms our hypothesis that binary classification leads to improved performance.

In the second step, we merged the messages that were labeled during the first step as false, true, partially false, and other to obtain a unified result. Messages without any label were assigned labels based on the basic mBERT classification results. In cases where a message received multiple labels, the label with the higher probability was selected. The classification results are presented in Table 10.

**Table 10.** Results of experiments with combined classes.

| Model | Accuracy | Macro F1-Score |
|---|---|---|
| mBERT | 0.35 | 0.26 |

We observe an improvement of 3% in macro F1-score compared to the basic multi-class classification. Detailed classification results are provided in Table 11.

**Table 11.** Detailed results of experiments with combined classes.

| Class | Precision | Recall | Macro F1-Score |
|---|---|---|---|
| False | 0.54 | 0.42 | 0.47 |
| True | 0.34 | 0.33 | 0.34 |
| Partially false | 0.13 | 0.11 | 0.12 |
| Other | 0.07 | 0.26 | 0.11 |

The results show that while the detection of false news remains consistent with the basic multi-class classification, there is an improvement of 1% for true news, 4% for partially false news, and a significant improvement of 8% (more than three times) for other news. These improvements demonstrate the effectiveness of the two-step classification approach and highlight its potential for future research.

## 7. Discussion

### 7.1. Results

In this paper, we explored various approaches for detecting and classifying fake news in case of multi-class classification. Our experiments were conducted on the CheckThat-2022 dataset using three transformer models: mBERT, SBERT, and XLM-RoBERTa. Among these models, mBERT yielded the best results with a macro F1-score of 0.23, although the state-of-the-art results on the dataset achieved a macro F1-score of 0.34. It is worth noting that all the results on the dataset were relatively low, likely due to the dataset's unbalanced nature, where the largest class (false news) is six times larger than the smallest class (other news).

To improve the performance of our models, we expanded the training CheckThat-2022 dataset by incorporating ChatGPT-generated news. Our experiments demonstrated that increasing the number of training messages led to improved results in multi-class classification. Notably, ChatGPT-generated messages showed high quality for true, partially true, and other news, enabling performance enhancements in classification for these classes. However, the generated false news did not contribute to the improvement of classification quality, which is likely due to ethical limitations in generating knowingly false information.

Furthermore, we introduced a novel two-step procedure for detecting partially false news. In the first step, we implemented four binary classifications, each highlighting a specific class against the remaining classes. This approach significantly improved the results of classification, particularly for the smaller classes, resulting in a 49% macro F1-score improvement for partially true news and a 46% improvement for other news.

In the second step, we combined the results obtained from the binary classifications to generate the final answer. This approach outperformed both the basic multi-class classification and the classification based on AI-generated news. These findings indicate the promising nature of our proposed method.

In conclusion, our study contributes to the field of fake news detection and classification by exploring various models, expanding datasets, and introducing a novel two-step classification approach. While there is room for further improvement, our results demonstrate the potential for enhancing the accuracy of multi-class fake news classification.

### 7.2. Limitations

Our work is subject to several limitations that deserve attention. Firstly, our obtained results demonstrate a considerable level of modesty. This can be attributed to three core factors: the multi-class classification nature, which represents a relatively novel approach within the realm of fake news classification, the unbalanced nature of the dataset, where the largest class (false news) is six times larger than the smallest class (other news), and the inherent challenge of dealing with diverse news topics present in the dataset. The latter point highlights that distinguishing news articles belonging to a single topic tends to be more straightforward.

It is worth noting that the observed suboptimal results are not a standalone challenge. As previously discussed, even the state-of-the-art (SOTA) results on the same dataset exhibit a similar pattern of limited success. This indicates that our endeavor is not isolated in its struggles, as other researchers face comparable issues. Yet, it is crucial to emphasize that without proactive efforts to advance results, progress is unlikely to manifest.

Secondly, while we delved into experiments involving combined classes, we exclusively employed the mBERT model for this purpose. To assert the superiority of our approach, a broader benchmarking with various models would be necessary. This entails comparing the application of mBERT in two distinct cases: with the usage of combined classes and with AI-generated texts. To extend our exploration of combined class utilization in the domain of fake news classification, a more comprehensive set of models should be enlisted.

## 8. Conclusions

In conclusion, our study effectively addressed our set objectives. By simulating real-world conditions characterized by diverse article topics and multi-class categorization, we conducted extensive experiments on a dataset encompassing four classification classes: false, true, partially false, and other news. Our investigation involved the fine-tuning of three promising transformer models—SBERT, XLM-RoBERTa, and mBERT—tailored to our specific research goals.

A significant contribution of our work is the introduction of the innovative concept of combined classes, a novel approach that improved the outcomes of multi-class classification. Furthermore, we leveraged ChatGPT to generate supplementary data, not only enhancing the balance and scope of our original dataset but also yielding improved classification results. This aspect demonstrated the viability of employing AI-generated text for data augmentation.

In advancing the realm of multi-class fake news detection, we not only expanded the theoretical framework but also presented practical strategies for enhancing the efficacy of multi-class fake news detection.

In the future, there are several interesting avenues for further improving the multi-class classification of fake news. Building upon the findings of this study, one promising direction is to expand the dataset by incorporating external sources. We have demonstrated that AI-generated news can enhance classification results, and it would be intriguing to explore alternative AI techniques beyond ChatGPT for generating different classes of news, particularly false news.

Another potential approach for performance improvement is to further explore the two-step classification methodology proposed in this paper. By rearranging the grouping of relevant classes (e.g., false, partially false vs. true, other), it may be possible to achieve even better classification results.

The concept of multi-class classification in the context of fake news detection holds great promise, as it closely mirrors real-life scenarios where news articles contain a mixture of truth and falsehood. To further refine the classification process, a future step could involve breaking down the partially false news class into more specific subclasses that capture the nuances and peculiarities of fake news. This could contribute to more effective and nuanced fake news detection algorithms.

Overall, the future of multi-class classification of fake news is ripe with opportunities for innovation and improvement. By expanding datasets, exploring alternative AI techniques, refining classification methodologies, and delving into more granular classification categories, we can strive towards more accurate and robust fake news detection systems.

**Author Contributions:** Conceptualization, E.S., M.A. and J.C.; Methodology, E.S., M.A. and J.C.; Software, E.S.; Resources, E.S.; Writing—original draft, E.S.; Writing—review & editing, E.S., M.A. and J.C.; Visualization, E.S.; Supervision, M.A. and J.C. All authors have read and agreed to the published version of the manuscript.

**Funding:** This research received no external funding.

**Data Availability Statement:** All datasets and codes utilized in this study are accessible at the following URL: https://doi.org/10.5281/zenodo.8195953 (accessed on 29 August 2023).

**Conflicts of Interest:** The authors declare no conflict of interest.

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
