# Peer review of "Improving Multiclass Classification of Fake News Using BERT-Based Models and ChatGPT-Augmented Data"

_inventions, doi:10.3390/inventions8050112_

Round 1

Reviewer 1 Report

The authors raise a very important and interesting topic related to the problem of fake news and the use of ChatGPT in this field.
The authors present a fairly short Introduction. However, taking into account the next chapter, the content of the Introduction is sufficient because it brings closer the problem and the reason why the authors took up this topic.
The chapter on the literature review has been divided into subchapters for easier review and, in conjunction with a fairly large amount of literature references, well shows the state of art.
Next, the experiment with the description of data sets is presented, supported by appropriate juxtapositions in the form of figures and tables. Although the study itself is not extensive, it is, in my opinion, valuable for the adopted collections and issues.
My only objection to the article itself, and not to the research work itself, is the lack of a separate Discussion chapter. If the authors, instead of interweaving their comments during the presentation of the results, compiled them appropriately, referring to their research and the already existing results of other teams, it would certainly be useful, and it would increase the value of the article itself. However, I do not insist, and even without this change, in my opinion, the article can be published. I leave the decisions to the editor.

Author Response

Thank you for your comprehensive review. In response to your feedback regarding "the lack of a separate Discussion chapter," we have included a dedicated Discussion section (please refer to page 11) that encompasses our thoughts on the results achieved and the limitations inherent in our work.

Reviewer 2 Report

1. Results: Recommend to be Major revisions      

This paper explores the potential of BERT-based models for detecting fake news in a diverse news corpus with unknown topics, encompassing four categories: true, false, partially false, and others. This investigation centers around the CheckThat! Lab dataset, which was introduced for the CLEF-2022 conference. Authors specifically examine the performance of the SBERT, RoBERTa, and mBERT models. To enhance the classification quality, we employ two methods: augmenting the training dataset with news generated by ChatGPT, and applying binary classification by combining classes and consolidating the results. The experimental results demonstrate superior performance compared to existing achievements, although the current state-of-the-art in FND still has room for improvement before practical applications can be realized.

This paper is with some merits for Inventions. However, it requires some major revisions.  

Firstly, the abstract should be refined to clearly indicate what authors had done within 120 words.

Secondly, for Sections 1 and 2, authors should provide the comments of the cited papers after introducing each relevant work. What readers require is, by convinced literature review, to understand the clear thinking/consideration why the proposed approach can reach more convinced results. This is the very contribution from authors. In addition, authors also should provide more sufficient critical literature review to indicate the drawbacks of existed approaches, then, well define the main stream of research direction, how did those previous studies perform? Employ which methodologies? Which problem still requires to be solved? Why is the proposed approach suitable to be used to solve the critical problem? We need more convinced literature reviews to indicate clearly the state-of-the-art development.

For Sections 3 to 5, authors should also introduce their proposed research framework more effective, i.e., some essential brief explanation vis-à-vis the text with a total research flowchart or framework diagram for each proposed algorithm to indicate how these employed models are working to receive the experimental results. It is difficult to understand how the proposed approaches are working.

For Section 6, authors should use more alternative models as the benchmarking models, authors should also conduct some statistical test to ensure the superiority of the proposed approach, i.e., how could authors ensure that their results are superior to others? Meanwhile, authors also have to provide some insight discussion of the results.

Moderate editing of English language required.

Author Response

Thank you for your insightful feedback. Based on your comments, we have made several notable improvements to the paper. Firstly, we have condensed the abstract to reduce its word count. Furthermore, we have enhanced the literature review section by incorporating more comprehensive details about relevant studies, and we've introduced a summarization at the end of this section (lines 181-186) to provide a cohesive overview.

To enhance the clarity of our methodologies, we've expanded our explanations of the principles underlying our approaches (Subsection 4.1 - Selection of BERT-based models) . We've also supplemented these explanations with visual aids in the form of figures to aid in understanding (new Figures 2-3).

Regarding Section 6, we have introduced a Limitation subsection within the paper (lines 390-409). This section outlines the shortcomings of our work, including the absence of a diverse array of models for benchmarking the experiments involving combined classes. We acknowledge that while this limitation exists, it is pertinent to note that our primary objective was to ascertain whether the application of combined classes enhances classification results compared to the state-of-the-art (SOTA) and the outcomes achieved via the utilization of the expanded dataset through ChatGPT. We recognize the importance of including other models for future benchmarking purposes.

Thank you once again for your thoughtful review, which has significantly contributed to the refinement of our paper.

Reviewer 3 Report

The article under review is devoted to the text analysis in order to detect false information in it. The work is of interest in the context of increasing data volumes.

However, there are a number of serious fundamental remarks over the article. There are some issues to be improved.

1. The detection of false information is highly dependent on the subject area - medicine, engineering and technology, economics, politics, etc. The article does not indicate in which knowledge space authors propose to detect false information.

2. The article proposes a classification into four classes - true, false, partially false, and other. However, there is no explanation what properties are characterized by each of these classes. Without such explanation, the further presentation of the text loses its meaning. How do the proposed classes differ from the generally accepted ones in machine learning, True Negatives (TN) and False Negative (FN) ?

3. The results of the classification given in the article, Table. 9-11 characterize the rather low algorithms performance. Based on the algorithms used, even improved ones, it is difficult to distinguish between potentially unreliable and truthful information.

4. There is no explanation what modifications are carried out by the authors with the BERT algorithm. In this regard, it is rather difficult to judge the contribution of the authors and the significance of the study in general.

Based on the foregoing, I believe that the article can be submitted for re-review after its significant revision.

Correction of spelling and punctuation inaccuracies required

Author Response

Thank you very much for your valuable feedback. We greatly appreciate your insights, which have prompted significant changes to the manuscript. In response to your comments, we have undertaken substantial revisions to enhance clarity and comprehensiveness.

Concerning your remark, "The detection of false information is highly dependent on the subject area”, in the list of our work's objectives (lines 33-46), we have now explicitly emphasized our aim to operate within real-life conditions. To achieve this, we selected a dataset that covers a wide array of topics, encompassing areas such as politics, health, crimes, environment, and more.

Concerning your remark, “The article proposes a classification into four classes - true, false, partially false, and other. However, there is no explanation what properties are characterized by each of these classes”, we have diligently elaborated on the classification categories, including false, true, partially false, and other news (lines 34-36, false (fake news), true (real news), partially false (articles that contains both false and true statements), and others (articles without clear classification evidence)). This information is now clearly expounded upon in the Dataset section (lines 194-199), complete with illustrative examples (Table 2).

In response to your feedback regarding “no explanation what modifications are carried out by the authors with the BERT algorithm”, in Subsection 4.1, "Selection of BERT-based Models" (lines 228-282), we have significantly expanded the content to thoroughly detail our modifications and approach for each of the mBERT, SBERT, and XLM-RoBERTa models.

Regarding the point you raised, “The results of the classification given in the article, Table. 9-11 characterize the rather low algorithms performance”, we have introduced a new Discussion section (please refer to page 11). Here, we meticulously analyse our results, acknowledging the modest performance we encountered. We recognize this as a shared challenge among researchers in similar pursuits and offer potential insights into the reasons behind these outcomes.

Concerning your remark, “it is rather difficult to judge the contribution of the authors and the significance of the study in general”, in the Conclusion section (please refer to page 12), we have explicitly outlined the key contributions of our work (the introduction of the innovative concept of combined classes, a novel approach that improved the outcomes of multi-class classification. Furthermore, we leveraged ChatGPT to generate supplementary data, not only enhancing the balance and scope of our original dataset but also yielding improved classification results. This aspect demonstrated the viability of employing AI-generated text for data augmentation. In advancing the realm of multi-class fake news detection, we not only expanded the theoretical framework but also presented practical strategies for enhancing the efficacy of multi-class fake news detection).

Once again, thank you for your constructive feedback, which has undoubtedly strengthened the quality and impact of our manuscript.

Reviewer 4 Report

The paper presents the potential of BERT-based models for detecting fake news in a diverse news corpus with unknown topics, encompassing four categories: true, false, partially false, and other. The Authors investigated the following BERT-based models: the SBERT, RoBERTa, and mBERT and employed two methods: augmenting the training dataset with news generated by ChatGPT, and applying binary classification by combining classes and consolidating the results. The experimental results were carried out. The paper aligns well with the journal's aims and scope.

The paper is well-structured, with a complete abstract and a clear statement of the authors' contributions at the end of the Introduction section. The literature review is complete.

Figure 1 should be provided in better quality – I suggest re-drawing it. Similarly – Figure 2.

In the Conclusions section, the authors should include limitations. Overall, the paper is good but requires some improvement in certain areas.

Author Response

We deeply appreciate your thorough review. Following your valuable feedback, we have included a new subsection titled "Limitations" (lines 390-409) where we delve into the challenges inherent to our work. This encompasses the modest results we encountered, a common experience among researchers tackling similar tasks, along with potential explanations for these outcomes. Additionally, we've taken steps to enhance the quality of our figures, thereby improving the overall visual presentation. Your insights have undoubtedly contributed to enhancing the comprehensiveness and quality of our manuscript.

Reviewer 5 Report

The topic is relevant and interesting.

The authors should present a list of contributions. The list of objectives can be use as a base. Write it in the present, not in the future.

Considering the multiclass classification, the

results are not relevant in a practical way,

but the topic is new and the authors well-explorenthe possibilities with a number of existing models.

The dataset expanded with chatgpt can be turns an overfitting, helping the models to find fake news. It is important to consider the analysis of it's impact.

After paper published it is important to spread opening the dataset and code.

Author Response

Thank you sincerely for your review. In response to your valuable feedback, we have outlined our contributions in the Conclusion section (lines 417-425). These contributions are built upon the objectives listed in the Introduction section and are supplemented by potential avenues for future exploration.

Round 2

Reviewer 2 Report

Authors have completely addressed all my concerns.

Moderate editing of English language required.

Reviewer 3 Report

The article has been amended in accordance with the reviewer comments. The article has been greatly improved. It can be recommended for publication.